# Euclasta condylotricha flowers essential oils: A new source of juvenile hormones and its larvicidal activity against Anopheles gambiae s.s. (Diptera: Culicidae)

**Roméo Barnabé Bohounton[1,2], Oswald Yédjinnavênan Djihinto[2], Oronce Sedjro-Ludolphe Dedome[1], Réné Mahudro Yovo[1], Laurette Djossou[2], Koffi Koba[3], Aristide Adomou[4], Pierre Villeneuve[5], Luc Salako Djogbénou[2,6], Fidèle Paul Tchobo○[1]***

1 Laboratory of Study and Research of Applied Chemistry, Polytechnic School of Abomey-Calavi, Cotonou, Benin, 2 Tropical Infectious Diseases Research Centre (TIDRC)/ University of Abomey Calavi, Abomey-Calavi, Benin, 3 Unité de Recherche sur les Matériaux et les Agroressources, École Supérieure D'agronomie, Université de Lomé, Lomé, Togo, 4 Laboratoire de Botanique et Écologie Végétale (LaBEV), Faculté des Sciences et Techniques (FAST), University of Abomey-Calavi, Abomey-Calavi, Benin, 5 Cirad, Persyst, UMR IATE, Montpellier, France, 6 Institut Régional de Santé Publique (IRSP), University of Abomey-Calavi, Ouidah, Benin

* fideletchobo@gmail.com

**Data Availability Statement:** All relevant data are within the paper and its Supporting Information files.

## Abstract

The essential oil (EO) of plants of the Poaceae family has diverse chemical constituents with several biological properties. But, data on the chemical constituents and toxicity are still unavailable for some species belonging to this family, such as Euclasta condylotricha Steud (Eu. condylotricha). In this study, the chemical composition of the EOs of Eu. condylotricha flowers was evaluated by gas chromatography coupled with mass spectrometry (GC–MS). The EOs larvicidal property was assessed against third instar larvae of three Anopheles gambiae laboratory strains (Kisumu, Acerkis and Kiskdr) according to the WHO standard protocol. The percentage yields of the EOs obtained from hydro distillation of Eu. condylotricha flowers varied 0.070 to 0.097%. Gas Chromatography-Mass Spectrometry (GC-MS) applied to the EOs revealed fifty-five (55) chemical constituents, representing 94.95% to 97.78% of the total essential oils. Although different chemical profiles of the dominant terpenes were observed for each sample, EOs were generally dominated by sesquiterpenoids with juvenile hormones as the major compounds. The primary compounds were juvenile hormone C16 (JH III) (35.97–48.72%), Methyl farnesoate 10,11-diol (18.56–28.73%), tau-Cadinol (18.54%), and β-Eudesmene (12.75–13.46%). Eu. condylotricha EOs showed a strong larvicidal activity with $LC_{50}$ values ranging from 35.21 to 52.34 ppm after 24 hours of exposition. This study showed that Eu. Condylotricha flowers essential oils are potent sources of juvenile hormones that could be a promising tool for developing an eco-friendly malaria vector control strategy.

**Funding:** This study received partial support from the Wellcome Trust in the form of an International Intermediate Fellowship in Public Health and Tropical Medicine grant to LSD [N° 109917/Z/15/Z]. The funders had no role in study design, data collection and analysis, publication decision, or manuscript preparation.

**Competing interests:** The authors have declared that no competing interests exist.

## Introduction

Essential oils (EOs), also called secondary metabolites, are volatile liquids with a strong odour obtained from various aromatic plant parts. EOs contain diverse bioactive compounds generated by plants to protect themselves against pathogens and insects [1, 2]. The bioactive compounds possess a wide range of biological activities such as toxicity and repellence to insects, antimicrobial, antioxidant, anticancer and antimalarial [3–6]. Since EOs are mixture of different bioactive compounds, those with proven larvicidal and adulticidal properties were assumed to offer less chance of resistance development in mosquito vectors and could be an alternative to synthetic insecticides [7].

In Benin, vector-borne infectious diseases, such as malaria, acquired through the bite of an infected female arthropod, are still a major public health concern. Mass distribution of insecticide-treated nets (ITNs) and indoor residual spraying (IRS) with chemical insecticides are the main strategies implemented in malaria vector control programs [8, 9]. Unfortunately, vectors resistance to the different classes of insecticides currently used in public health (pyrethroids, carbamates, and organophosphates) could jeopardize the vector control interventions [10]. Indeed, several studies have reported that natural mosquito populations in Benin showed resistance to pyrethroids, DDT, carbamates and organophosphates insecticides [11, 12]. In addition, N'guessan et al. [13] have demonstrated that, pyrethroid resistance in the primary malaria vector *An. gambiae* was threatening the effectiveness of ITN and IRS in the areas of high resistance in Benin. The widespread of the insecticide resistance phenomenon in addition to the environmental pollution from chemical insecticides and its high operational cost have led to the need to develop alternative malaria control approaches. Developing effective and eco-friendly tools for reducing the burden or eliminating malaria is highly recommended in this context.

The plant extracts properties as insecticide, repellent or fumigant against mosquitoes and other pests and insect vectors have been well reported. Especially, EOs of several plants have been shown to exhibit significant insecticidal and repellent properties against adult mosquitoes [14–16], as well as against mosquito larvae [17–20]. Therefore, it is worth investigating the Beninese flora to find out the plant species whose essential oils could display insecticidal activity against malaria vectors. The grass genus *Euclasta* Franch (Poaceae) comprises two species, *Euclasta clarkei* (Hack.) Cope which is distributed in South-western Asia, and *Euclasta condylitricha* (Steud) (*Eu. condolytricha*) which is found in Africa, southern Asia and America [21]. *Eu. condolytricha*, also known as *Andropogon condylotrichus*, is an annual species, and the height of mature plants can reach up to 2 meters [22]. To our knowledge, there is no report on the chemical constituents of *Eu. condyloticha* essential oil nor its insecticidal activity. Considering the concerted efforts to develop plant products-based insecticides as an excellent alternative to synthetic insecticides, the current study aims to: (i) investigate the chemical profile of the essential oils obtained from *Eu. condyloticha* flowers harvested in Benin; (ii) evaluate the larvicidal activity of the EOs on *Anopheles gambiae* s.s. mosquitoes.

## Experimental

### Plant material and extraction

*Euclasta condyloticha* Steud flowers (Fig 1) were collected in November 2018 from the localities of Ouessè (middle) (8°38'19.3"N, 2°38'04.7"E), Parakou (northeast) (9°22'55.2"N, 2°36'48.1"E) and Sinendé (north-west) (10°06'52.7"N, 2°22'58.7"E) of Benin Republic. The taxonomic identification of the plants was carried out at the 'Laboratoire de Botanique et Écologie Végétale (LaBEV) of the University of Abomey-Calavi, Benin, and the voucher specimens are

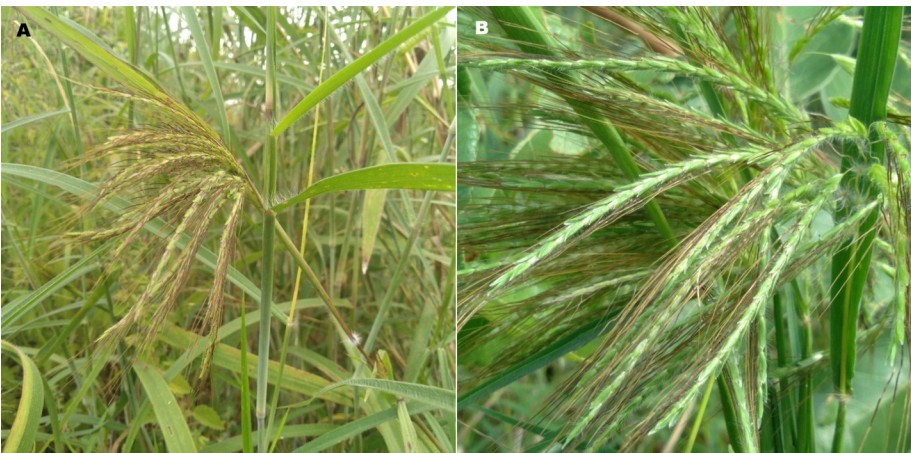

**Fig 1. *Euclasta condylotricha* Steud harvested in Benin. A**: The whole plant with stem, leaves and flowers. **B**: Areal part of the plant showing the flowers.

kept at the National Herbarium of the University. The voucher specimen numbers are AAC 190/HNB, AAC 191/HNB and AAC 192/HNB respectively for samples from Ouessè, Parakou and Sinendé.

The collected flowers were dried at 25°C ±2°C for 72 hours. Three batches of 100 g of dried flowers were submitted to hydro-distillation in the Clevenger apparatus at 100°C for 150 min. The distilled oil was dried using anhydrous sodium sulphate, transferred into an airtight amber-colored vial, and stored at 4°C until further use. The yields were averaged over the three extraction experiments per plant material.

## Analysis of volatile compounds

The chemical composition of the EOs was evaluated by gas chromatography coupled with mass spectrometry (GC–MS) according to a methodology previously reported [23]. Compounds identifications were performed using published data [24] and a comparison with the NIST mass spectral library.

## Mosquito strains

Three *An. gambiae* s.s. laboratory strains (Kisumu, Acerkis and Kiskdr) from the insectary of the laboratory of Vector-Borne Infectious Diseases at the Institut Régional de Santé Publique Alfred Quenum (IRSP-AQ) of the University of Abomey-Calavi in Ouidah (Benin) were used in this study. Kisumu strain originating from Kenya is a reference strain susceptible to all insecticides [25]. Acerkis, resistant to organophosphate and carbamate-based insecticides, is homozygous for (G119S) mutation [26]. Kiskdr is homozygous for $kdr^R$ (L1014F) allele and confers resistance to pyrethroids and DDT [27]. Both AcerKis and Kiskdr were supposed to share the same genetic background as the Kisumu strain but differ by the presence of resistance alleles. The colonies of the three strains were maintained in the insectary under optimum conditions (25–27°C temperature, 70–80% relative humidity and 12:12 light and dark period).

## Larval bioassay

Third instar larvae of each mosquito strain were used for the bioassays. Bioassays were performed according to the standard method recommended by the World Health Organization

as previously described in [23], using seven concentrations of each EO sample as follow: 10, 20, 30, 40, 50, 60 and 70 ppm.

## Data analysis

The analysis of dose-mortality responses in larval bioassays was performed using the BioRssay script version 6.2 [32] in R software Version 3.0 [33]. This script calculates the mortality-dose regression using a generalised linear model (GLM). To assess the adequacy of the model, a chi-square test between the observed dead numbers and the dead numbers predicted by the regression is used. Using a likelihood ratio test (LRT), it also tests whether the mortality-dose regressions are similar for the different strains, using a likelihood ratio test (LRT). If there are more than two strains test, it also computes the pairwise test and corrects it using sequential Bonferroni correction [28]. Finally, it computes the lethal concentrations inducing 50% ($LC_{50}$) and 95% ($LC_{95}$) mortality recorded in each mosquito strain and the associated confidence intervals; the resistance ratios, i.e. $RR_{50}$ or $RR_{95}$ ($LC_{50}$ or $LC_{95}$ in each strain, divided respectively by the $LC_{50}$ or $LC_{95}$ of the reference strain) and their 95% confidence intervals. Susceptible or resistant status was defined according to Mazzarri & Georghiou [29] and Bisset et *al.* [30] criteria: $RR_{50} \leq 1$ and $RR_{50} > 1$ indicate respectively susceptibility and phenotypic resistance response against the essential oil.

## Results and discussion

### Essential oils chemical composition

This study characterised the EOs chemical composition of *Eu. condylotricha* flowers harvested in three localities (Ouessè, Parakou and Sinendé) of the Benin Republic. The dried flowers of *Eu. condylotricha* were hydrodistilled and produced an EOs with a characteristic odour. The extractions yields were 0.097%, 0.070% and 0.085%, respectively for EOs from Ouessè, Parakou and Sinendé plant samples.

Gas Chromatography-Mass Spectrometry (GC-MS) applied to the EOs revealed fifty-five (55) chemical constituents, representing 94.95% to 97.78% of the total essential oils (Table 1). However, four compounds have not been identified, including one compound common to the three EO samples (1.27 to 2.12%) and three compounds in EO from Sinendé (0.31% to 1.69%). The EO sample from Sinendé contained forty-four (44) compounds representing 95.86% of the oil. Forty (40) chemical constituents were identified in the EO from Parakou, accounting for 94.95% of the essential oil's mass. The lowest number of phytochemicals, thirty-five (35) components, representing 95.86% of all the compounds, was recorded in the EO from Ouessè. There was a wide range of variations in EOs constituents. Each of the three EOs was dominated by juvenile hormones. Indeed, the primary class of components identified was juvenile hormones with 74.87%, 61.29% and 54.92%, respectively in EOs samples from Ouessè, Parakou and Sinendé. Among the juvenile hormones, the most abundant chemical constituents were the juvenile hormone C16 (48.72% in EO from Ouessè; 35.97% in EO from Sinendé; and 32.08% in EO from Parakou), followed by methyl farnesoate 10,11-diol (28.73%, 25.67%, and 18.56% respectively in EO from Parakou, Ouessè, and Sinendé) (Table 1). The sesquiterpenes were the second representative class of components identified in three EOs samples. The major group of sesquiterpenes was the hydrocarbons sesquiterpenes (15.17% in Ouessè, 27.77% in Parakou and 15.36% in Sinendé), while the oxygenated sesquiterpenes accounted for 1.66% in Ouessè, 2.23% in Parakou and 21.44% in Sinendé. The hydrocarbons sesquiterpene β-eudesmene was found in EO from Parakou (24.12%), Ouessè (13.36%), and Sinendé (12.75%). Among the oxygenated sesquiterpenes, the major compound was Tau-cadinol (18.54%), found mainly in the EO from Sinendé (Table 1).

**Table 1. Chemical composition of the *Euclasta condylotricha* Steud essential oils.**

| No | Rt | Compounds | Percentage of total composition | | |
|---|---|---|---|---|---|
| | | | Ouessè | Parakou | Sinendé |
| 1 | 13.87 | α-Pinene | - | tr | - |
| 2 | 15.08 | β-Myrcene | tr | tr | tr |
| 3 | 17.08 | o-Cymene | - | tr | - |
| 4 | 17.36 | β-Terpinyl acetate | - | tr | - |
| 5 | 22.5 | Linalool | - | 0.15 | tr |
| 6 | 26.76 | trans-Borneol | tr | - | tr |
| 7 | 26.8 | 3-Pinanylamine | - | - | tr |
| 8 | 27.6 | Terpinen-4-ol | tr | tr | - |
| 9 | 28.59 | L-α-Terpineol | - | tr | - |
| 10 | 28.93 | n-Hexyl butanoate | tr | tr | tr |
| 11 | 29.2 | trans-2-Hexenyl Butyrate | tr | tr | tr |
| 12 | 32.97 | 2-Methylbutyl caproate | tr | tr | tr |
| 13 | 41.67 | (-)-β-Elemene | 0.72 | 1.15 | 1.02 |
| 14 | 41.93 | (2Z)-2-Hexenyl butyrate | tr | 0.15 | - |
| 15 | 42.73 | β-Gurjunene | tr | tr | - |
| 16 | 43.19 | β-Caryophyllene | 0.36 | 0.44 | 0.33 |
| 17 | 44.5 | L-Alloaromadendrene | tr | tr | tr |
| 18 | 45.28 | α-Humulene | 0.17 | 0.17 | 0.23 |
| 19 | 45.43 | E.E-8.10-Dodecadien-1-ol | - | 0.22 | tr |
| 20 | 45.52 | α-Selinene | 0.16 | 0.15 | 0.15 |
| 21 | 46.08 | 1.6-Dimethylhepta-1.3.5-triene | 1.03 | 0.57 | 0.84 |
| 22 | 46.73 | (+)-Aromadendrene | 0.11 | 0.20 | 0.15 |
| 23 | **47.32** | **β-Eudesmene** | **13.46** | **24.12** | **12.75** |
| 24 | 47.83 | γ-Gurjunene | 0.94 | 2.45 | 1.18 |
| 25 | 48.4 | Aciphyllene | 0.13 | 0.25 | 0.16 |
| 26 | 49 | γ-Cadinene | - | - | 0.56 |
| 27 | 49.12 | (-)-α-Panasinsene | tr | 0.14 | tr |
| 28 | 49.63 | NI | - | - | 0.31 |
| 29 | 50.38 | α-Cadinene | - | - | tr |
| 30 | 51.15 | Elemol | 0.13 | tr | 0.10 |
| 31 | 52.02 | Geranyl isobutyrate | - | 0.10 | - |
| 32 | 52.12 | D-Nerolidol | 0.11 | - | 0.26 |
| 33 | 52.53 | β-copaene | - | - | tr |
| 34 | 52.63 | (+)-Spathulenol | tr | tr | - |
| 35 | 52.87 | β-Caryophyllene oxide | tr | tr | tr |
| 36 | 53.41 | Epiglobulol | - | - | 0.42 |
| 37 | 53.88 | Ledol | tr | tr | tr |
| 38 | 54.66 | (-)-Neointermedeol | 0.34 | 0.61 | 0.28 |
| 39 | 54.79 | Epicubenol | - | - | 0.23 |
| 40 | 55.10 | 7-epi-cis-sesquisabinene hydrate | - | tr | - |
| 41 | 55.12 | Germacrene D-4-ol | - | - | tr |
| 42 | 55.54 | Di-epi-1.10-cubenol | - | - | tr |
| 43 | 55.74 | (-)-β-Longipinene | tr | tr | tr |
| 44 | **56.36** | **tau.-Cadinol** | - | - | **18.54** |
| 45 | 56.67 | β-Eudesmol | tr | 0.10 | - |
| 46 | 56.98 | Globulol | 0.52 | 0.96 | 1.08 |

(*Continued*)

**Table 1.** (Continued)

| No | Rt | Compounds | Percentage of total composition | | |
|---|---|---|---|---|---|
| | | | Ouessè | Parakou | Sinendé |
| 47 | 58.89 | Shyobunol | - | - | tr |
| 48 | 62.59 | NI | - | - | 0.49 |
| 49 | **64.27** | **Methyl farnesoate 10.11-diol** | **25.67** | **28.73** | **18.56** |
| 50 | 66.58 | 2.3-Dimethylpentanal | tr | tr | tr |
| 51 | 66.99 | Juvenile hormone C18 | 0.48 | 0.48 | 0.39 |
| 52 | **68.36** | **Juvenile hormone C16** | **48.72** | **32.08** | **35.97** |
| 53 | 68.55 | NI | 2.12 | 1.27 | 1.56 |
| 54 | 69.23 | NI | - | - | 1.69 |
| 55 | 69.93 | trans-Z-α-Bisabolene epoxide | 0.69 | 0.46 | 0.53 |
| | | **Oxygenated monoterpenes** | **0.54** | **1.3** | **0.9** |
| | | **Sesquiterpene hydrocarbons** | **15.17** | **27.77** | **15.36** |
| | | **Oxygenated sesquiterpenes** | **1.66** | **2.23** | **21.44** |
| | | **Juvenile hormones** | **74.87** | **61.29** | **54.92** |
| | | **Others** | **3.62** | **2.49** | **5.29** |
| | | **Total identified (%)** | **95.86** | **94.95** | **97.78** |

Compounds are listed in order of elution from a HP- 5MS column. **Rt**: retention times; **tr**: trace amount (<0.1%); (-): not detected; **NI**: not identified.

The EOs analysed in this study were enriched in juvenile hormones. These compounds were known as insect hormones involved in vital physiological processes in larvae and adult insects [31, 32]. However, to date, only two studies reported the presence of juvenile hormones in plant extracts. The first by Toong *et al.*, [33] in the whole plants of *Cyperus iria* L and *C. aromaticus* Ridl. These authors showed that the excess of the juvenile hormone could distort the wings, change the colour, and induce infertility in adult migratory grasshopper *Melanoplus sanguinipes* females. The second study has shown the existence of juvenile hormone in aqueous methanolic extract of *Cananga latifolia* stem bark [34]. Moreover, juvenile hormones were reported to affect the development of nematodes and represent valuable phytocompounds for insect pest control [31, 35].

We observed a significant variation in the yield and composition of the essential oils of *Eu. condylotricha* flowers collected in three different localities in the Benin Republic. This variation can be attributed to the different environmental conditions, such as altitude, solar exposure, and soil composition [36]. Such variations in EOs yield and phytochemical constituents dependent on the plant's geographical area have been reported for several other plant species, demonstrating that distinct plants from different locations display different chemotypes [37, 38].

The current study is the first work on *Eu. condylotricha* (Poaceae) flowers' EOs composition. However, several studies have been carried out on the chemical composition of essential oils from other plants of the Poaceae family. Indeed, it was shown that the essential oils of *Bothriochloa* spp.; *Cymbopogon* spp. and *Chrysopogon zizanioides* are dominated by the presence of sesquiterpenes [39–42]. As sesquiterpenes have been found as major constituents of the EOs analysed in this study, EOs from the Poaceae family is enriched in sesquiterpenes.

## Larvicidal activity

For several decades, plant derivatives like EOs from aromatic plants have been screened for effective larvicidal activities against *An. gambiae* larvae [43–45]. The EOs of *Eu. condylotricha*

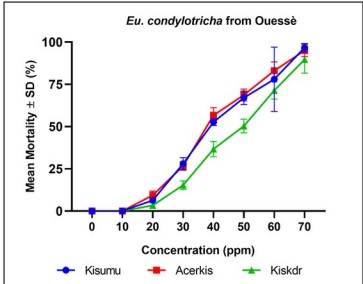 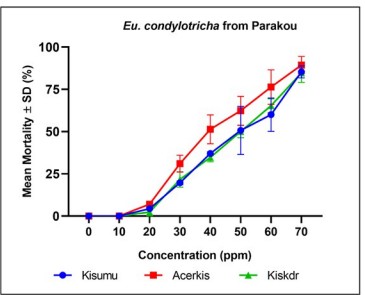 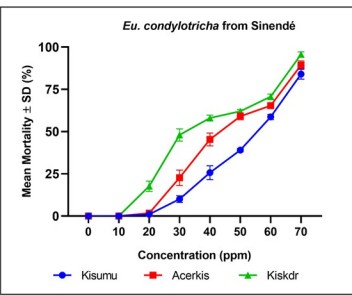

**Fig 2. Percentage of dead *An. gambiae* larvae after 24 h exposition at different concentrations of EOs.** 0 ppm indicates control solution with water and ethanol without EO.

flowers were tested against third instar larvae of two resistant (Kiskdr, Acerkis) and one susceptible (Kisumu) laboratory strains of *An. gambiae*.

The larvicidal activity of the EOs under a series of concentrations ranging from 10 to 70 ppm were evaluated. As shown in Fig 2, toxicity of the three EOs is concentration-dependent. The EOs exhibited significant larvicidal activity, with $LC_{50}$ values ranging from 35.21 to 52.34 ppm (Table 2) after 24 hours of exposure. The EO sample from Sinendé was the most active, with $LC_{50}$ of 35.21 ppm on Kiskdr larvae, followed by the EO from Ouessè with $LC_{50}$ of 38.10 ppm and 38.46 ppm on Acerkis and Kisumu larvae, respectively.

Acerkis strain larvae ($LC_{50}$ = 38.10 ppm) were significantly more susceptible to EO from Ouessè compared to Kiskdr ($LC_{50}$ = 46.30 ppm, $p < 0.001$), while no significant difference was observed compared to Kisumu larvae ($LC_{50}$ = 38.46 ppm, $p = 0.33$). However, for the Sinendé sample, Kisumu and Kiskdr larvae susceptibility was significantly different ($p < 0.001$). The $LC_{50}$ value recorded in Kiskdr larvae (35.21 ppm) was significantly lower than that in Acerkis (44.45 ppm, $p < 0.001$) and Kisumu larvae (52.34 ppm, $p < 0.001$).

This is the first report on the larvicidal activity of *Eu. condylotricha* flowers OEs against *An. gambiae* larvae. In agreement with the criteria established by Cheng *et al*., [46] plant EOs showing $LC_{50}$ values $< 50$ ppm within 24 hours were very active, and $LC_{50}$ (24h) values $< 100$ ppm were declared active. Based on those classifications, *Eu. condylotricha* flowers EOs could represent an inexpensive and environmentally benign agent for controlling malaria vectors.

**Table 2. Toxicity against *An. gambiae* larvae after 24 h exposure.**

| EOs samples | Strains | LC$_{50}$ (ppm) | 95% C.I | RR$_{50}$ | 95% CI [LCL-UCL] | LC$_{95}$ (ppm) | 95% CI [LCL-UCL] | Slope ±S.E | Intercept ±S.E | Chi(p) value |
|---|---|---|---|---|---|---|---|---|---|---|
| | Kisumu | 38.46 | 36.94–39.96 | - | - | 74.45 | 69.65–80.67 | 5.73 ± 0.45 | -9.09 ± 0.47 | 0.98 |
| **Ouessè** | Acerkis | 38.10 | 36.43–39.75 | 0.99 | 0.86–1.14 | 79.33 | 73.42–87.20 | 5.16 ± 0.28 | -8.16 ± 0.45 | 0.95 |
| | Kiskdr | 46.30 | 44.15–48.59 | 1.20 | 1.04–1.38 | 92.05 | 83.43–104.65 | 5.51 ± 0.38 | -9.18 ± 0.63 | 0.71 |
| | Kisumu | 48.06 | 45.99–50.31 | - | - | 108.81 | 98.07–124.02 | 4.63 ± 0.27 | -7.79 ± 0.45 | 0.92 |
| **Parakou** | Acerkis | 40.25 | 38.09–42.55 | 0.84 | 0.73–0.95 | 92.20 | 82.94–105.51 | 4.57 ± 0.29 | -7.33 ± 0.48 | 0.64 |
| | Kiskdr | 47.41 | 45.36–49.60 | 0.99 | 0.87–1.21 | 102.88 | 93.09–116.72 | 4.88 ± 0.29 | -8.19 ± 0.49 | 0.59 |
| | Kisumu | 52.34 | 50.36–54.50 | - | - | 100.72 | 92.10–112.85 | 5.78 ± 0.35 | -9.94 ± 0.6 | 0.69 |
| **Sinendé** | Acerkis | 44.45 | 42.23–46.79 | 0.85 | 0.74–0.97 | 93.49 | 84.18–107.20 | 5.09 ± 0.35 | -8.4 ± 0.58 | 0.45 |
| | Kiskdr | 35.21 | 31.61–38.63 | 0.67 | 0.59–0.77 | 103.31 | 85.28–138.55 | 3.51 ± 0.38 | -5.44 ± 0.61 | 0.44 |

No mortality was observed in the control

LC$_{50/95}$: lethal concentrations; **S.E**: standard error; **C.I**: Confidence interval; **RR$_{50}$** is resistance ratio at LC$_{50}$: LC$_{50}$ (resistant strain)/ LC$_{50}$ (Kisumu). **LCL**: Lower confidence limit; **UCL**: Upper confidence limit

Chi(p) is indicated to judge whether the data are well fitted to the regression or not. The fits are acceptable when the p-value is over 0.05.

The $RR_{50} < 1$ suggests that the pyrethroid-resistant (Kiskdr) and the carbamate/organophosphate resistant (Acerkis) mosquito strains were susceptible to the *Eu. condylotricha* flowers essential oils, except Kiskdr strain to the sample from Ouessè. This finding indicates that the presence of insecticide resistance mutations (L1014F in Kiskdr and G119S in Acerkis) did not induce a cross-resistance to the EO extracts. Although, there were no previous studies on the larvicidal activity of *Eu. condylotricha* flowers EO, some EOs from the plants of the Poaceae family with proven bioactivity against *Anopheles* species larvae have been reported [47–49]. The larvicidal effect of *Eu. condylotricha* EOs might be caused by the secondary metabolites contained in the OEs. The high amount of juvenile hormones and other sesquiterpenes could be the basis of the observed larvicidal activity. Indeed, several ethnobotanical studies have reported that many sesquiterpene-rich EOs demonstrated excellent larvicidal activities. This is the case of *Murraya exotica* EO with 61.5% of sesquiterpene hydrocarbons and 6.01% of oxygenated sesquiterpenes that exhibited strong larvicidal activity against *Anopheles stephensi* larvae ($LC_{50} = 31.3$ ppm) [50]; *Zingiber nimmoni* EO that contains 51.9% of sesquiterpene hydrocarbons and 16.2% of oxygenated sesquiterpenes which showed high larvicidal activity against *Anopheles stephensi* larvae ($LC_{50} = 41.2$ ppm) [51]; and *Chloroxylon swietenia* EO that has 53.9% of sesquiterpene hydrocarbons and 3.0% of oxygenated sesquiterpenes which was very active on *Anopheles stephensi* larvae ($LC_{50} = 14.9$ ppm) [52]. Moreover, some sesquiterpenes exhibited larvicidal activity against mosquito larvae. Sesquiterpenes such as humulene ($LC_{50} = 6.19$ ppm); caryophylene ($LC_{50} = 41.66$ ppm); elemene ($LC_{50} = 10.26$ ppm); germacrene D-4-ol ($LC_{50} = 6.12$ ppm) and $\alpha$-cadinol ($LC_{50} = 10.27$ ppm) were demonstrated to be toxic to *Anopheles subpictus* larvae [51, 53]. Caryophyllene oxide ($LC_{50} = 49.81$ ppm) and germacrene D ($LC_{50} = 49.46$ ppm) were also shown toxic to *Anopheles anthropophagus* larvae [54].

## Conclusion

Local plant derivatives with insecticidal properties constitute a promising alternative for malaria vector control. The EOs of *Eu. condylotricha* samples collected in different regions of Benin Republic were analyzed and found to contain high concentrations of sesquiterpenoid compounds. The juvenile hormone C16 was the major constituent of the phytochemical compounds of all EOs samples. The first report of this hormone in *Eu. condylotricha* provides solid background for a wide range of applications for malaria control. The OEs showed significant larvicidal activity against resistant *An. gambiae* strains. Before translating these research findings into operational interventions, further investigations on mechanisms by which EOs mediate their bioinsecticidal activity may require evaluations of the major EOs' constituents separately. This study findings contribute to the dissemination of knowledge regarding the chemical composition and larvicidal activity of this Beninese plant species, which is almost incipient in the literature.

## Supporting information

**S1 File. Larval bioassay raw data.**
(XLSX)

**S2 File. Representative GC–MS chromatogram of *Eu. Condylotricha* flowers essential oils.**
(DOCX)

## Author Contributions

**Conceptualization:** Roméo Barnabé Bohounton, Pierre Villeneuve, Luc Salako Djogbénou, Fidèle Paul Tchobo.

**Data curation:** Roméo Barnabé Bohounton, Oswald Yédjinnavênan Djihinto, Oronce Sedjro-Ludolphe Dedome, Réné Mahudro Yovo, Laurette Djossou, Koffi Koba, Aristide Adomou.

**Formal analysis:** Roméo Barnabé Bohounton.

**Supervision:** Fidèle Paul Tchobo.

**Writing – original draft:** Roméo Barnabé Bohounton.

**Writing – review & editing:** Roméo Barnabé Bohounton, Oswald Yédjinnavênan Djihinto, Oronce Sedjro-Ludolphe Dedome, Luc Salako Djogbénou, Fidèle Paul Tchobo.

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
