## [Decision Letter · Decision Letter 0]

10 Oct 2022

PONE-D-22-18507

Euclasta condylotricha flowers essential oils: a new source of juvenile hormones and its larvicidal activity against Anopheles gambiae s.s. (Diptera: Culicidae)

PLOS ONE

Dear Dr. Tchobo,

Thank you for submitting your manuscript to PLOS ONE. After careful consideration, we feel that it has merit but does not fully meet PLOS ONE’s publication criteria as it currently stands. Therefore, we invite you to submit a revised version of the manuscript that addresses the points raised during the review process.

We look forward to receiving your revised manuscript.

Kind regards,

Guadalupe Virginia Nevárez-Moorillón, Ph.D.

Academic Editor

PLOS ONE

Journal Requirements:

2. Please describe any permissions and permits you had to collect plant samples.

   "RBB received financial support through Wellcome Trust intermediate fellowship in public health and tropical medicine grant (N° 109917/Z/15/Z) awarded to LSD. The funders had no role in study design, data collection and analysis, decision to publish, or preparation of the manuscript."

Additional Editor Comments:

Please, take in consideration the suggestions done by the reviewers. 

Reviewers' comments:

Reviewer's Responses to Questions

**Comments to the Author**

1. Is the manuscript technically sound, and do the data support the conclusions?

Reviewer #1: Partly

Reviewer #2: Yes

2. Has the statistical analysis been performed appropriately and rigorously? 

Reviewer #1: Yes

Reviewer #2: Yes

3. Have the authors made all data underlying the findings in their manuscript fully available?

Reviewer #1: Yes

Reviewer #2: Yes

4. Is the manuscript presented in an intelligible fashion and written in standard English?

Reviewer #1: Yes

Reviewer #2: Yes

5. Review Comments to the Author

Reviewer #1: The paper “Euclasta condylotricha flowers essential oils: a new source of juvenile hormones and its larvicidal activity against Anopheles gambiae s.s. (Diptera: Culicidae)”, describes the extraction of essential oils from flowers of a grass and the chemical identification of its components by gas chromatography coupled to mass spectrometry. In addition, the larvicidal activity of essential oil was evaluated in Anopeles gambiae mosquitoes. Authors found JH III as the main compound in the essential oil. The essential oil showed strong larvicidal activity and authors propose the flowers from these grass as a source of juvenile hormone than can be used as mosquito control strategy. The results are interesting, in the light of control of vector-borne diseases. This reviewer have few concerns that should be addressed before accepted for publication.

Lines 76-77. “Considering the concerted efforts to develop plant products-based insecticides as an excellent alternative to synthetic insecticides,..” It is an interesting idea that has been proposed long time ago, the question is, how much (or how we evaluate) non-target species could be affected in their habitats by the use of such strategies?

Line 169. “…whole plants of Cyperus iria L.”, should say “…whole plants of Cyperus iria L. and C. aromaticus Ridl.”

Line 184. “…EOs…” instead “…essential oils”

Lines 245-247. Clearly, results show that EOs have a larvicidal effect, but there is not a direct prove that such effect was by JH. It is only an association because our knowledge that JH III interfere with larvae development, but in EOs more components are present and no evaluations separately were performed. Conclusion (and/or discussion) should be clear on those limitations.

Reviewer #2: In this paper, the authors present a well-structured work on the characterization and larvicidal effect of Euclasta condylotricha against Anopheles gambiae with promising results against the malaria vector.

The results are clear and well represented, I have only minor suggestions to support the publication of the paper.

Line 73: Proper spelling of the scientific name "Andropogon condylotrichus" remove the capital letter of the species.

Line 195: The value 35.54 does not appear in table 2; correct for the correct data.

Line 197-198: It should be: with LC50 of 38.10 ppm and 198 38.46 ppm....

Figure 2. The title should be "Percentage of dead An. gambiae larvae..." instead of "Number of dead An. gambiae larvae...". In the figures, the graphs and text are not clear, I suggest that they be replaced by figures of higher image quality.

6. PLOS authors have the option to publish the peer review history of their article (what does this mean?). If published, this will include your full peer review and any attached files.

Reviewer #1: No

Reviewer #2: No

---

## [Author Response · Author response to Decision Letter 0]

2 Nov 2022

Dear Editor,

Please find below the responses addressing each of the comments of the reviewers assigned to our submitted manuscript entitled: Euclasta condylotricha flowers essential oils: a new source of juvenile hormones and its larvicidal activity against Anopheles gambiae s.s. (Diptera: Culicidae).

Editor

Reply: We reviewed again the journal formatting guidelines and ensured that the manuscript meets the style requirements.

2. Please describe any permissions and permits you had to collect plant samples.

Reply: Please, kindly note that at the University of Abomey-Calavi, there is not yet a policy to deliver permission or permit before plant collection. The general rule at the University is that after plant collection, the author deposits a plant sample at the National Herbarium and gets a voucher number.

Furthermore, the plant species used in this study was collected in the field in its natural habitat. The plant is an annual species, very wide distributed that colonizes agricultural lands and vacant fields. In that case, no permission or permit is delivered prior to the plant samples collection.

3. Thank you for stating the following financial disclosure: "RBB received financial support through Wellcome Trust intermediate fellowship in public health and tropical medicine grant (N° 109917/Z/15/Z) awarded to LSD. The funders had no role in study design, data collection and analysis, decision to publish, or preparation of the manuscript."

Reply: RBB received material support through the grant awarded to LSD. No grants or organizations supported the study. No funding was received from our institution.

Reply: The funders had no role in study design, data collection and analysis, decision to publish, or preparation of the manuscript.

Reply: No author received a salary from the funder.

Reply: The authors received no specific funding for this work.

Reply: Amended statements are provided in the cover letter.

4. PLOS requires an ORCID iD for the corresponding author in Editorial Manager on papers submitted after December 6th, 2016. Please ensure that you have an ORCID iD and that it is validated in Editorial Manager. To do this, go to ‘Update my Information’ (in the upper left-hand corner of the main menu), and click on the Fetch/Validate link next to the ORCID field. This will take you to the ORCID site and allow you to create a new iD or authenticate a pre-existing iD in Editorial Manager. Please see the following video for instructions on linking an ORCID iD to your Editorial Manager account: https://www.youtube.com/watch?v=_xcclfuvtxQ.

Reply: ORCID detail of the corresponding author was provided and validated in the Editorial Manager.

Reply: Done.

Reviewer #1

The authors thank the reviewer for his comments on the manuscript.

Lines 76-77. “Considering the concerted efforts to develop plant products-based insecticides as an excellent alternative to synthetic insecticides,..” It is an interesting idea that has been proposed long time ago, the question is, how much (or how we evaluate) non-target species could be affected in their habitats by the use of such strategies?

Reply: This is a question that needs further investigations before implementing such strategy base on plant products. The final formulation of the plant-based products to be applied to the larval habitats need to be determined. The choice of an appropriate formulation will minimize the non-target impacts. To evaluate the effect of such strategy on non-target species, the final formulation to be delivered in the field should assess the toxicological effects of treating representative, sensitive indicator species, such as an aquatic invertebrates and vertebrates with the plant product. 

Furthermore, since plant-based products are natural and eco-friendly materials, we assume that these products has no greater toxicological effects than that from the actual chemical larvicides.

Line 169. “…whole plants of Cyperus iria L.”, should say “…whole plants of Cyperus iria L. and C. aromaticus Ridl.”

Reply: C. aromaticus Ridl was added at the end of the sentence. Done.

Line 184. “…EOs…” instead “…essential oils”

Reply: Done

Lines 245-247. Clearly, results show that EOs have a larvicidal effect, but there is not a direct prove that such effect was by JH. It is only an association because our knowledge that JH III interfere with larvae development, but in EOs more components are present and no evaluations separately were performed. Conclusion (and/or discussion) should be clear on those limitations.

Reply: Please, notice that in Lines 245-247, we did not mean that the larvicidal effect of the EOs is due to the Juvenile Hormone. As conclusion, we were reporting the major phytochemical compound found in all EOs. In the discussion session, lines 225-241, we stated that the larvicidal effect could be due to the secondary metabolites present in the EOs (Juvenile hormone and sesquiterpenes). However, we added a statement in the conclusion. Please see line 249 to 252 as follow: 

“…Before the translation of these research findings into operational interventions, further investigations on mechanisms by which EOs mediate their bioinsecticidal activity may require evaluations of the major EOs’ constituents separately…”

Reviewer #2: 

We are grateful to the reviewer for his comments on the manuscript.

Line 73: Proper spelling of the scientific name "Andropogon condylotrichus" remove the capital letter of the species.

Reply: Done.

Line 195: The value 35.54 does not appear in table 2; correct for the correct data.

Reply: Done.

Line 197-198: It should be: with LC50 of 38.10 ppm and 198 38.46 ppm....

Reply: Done.

Figure 2. The title should be "Percentage of dead An. gambiae larvae..." instead of "Number of dead An. gambiae larvae...". In the figures, the graphs and text are not clear, I suggest that they be replaced by figures of higher image quality.

Reply: The Title of Figure 2 was changed accordingly. The low quality of the Figure 2 may be due to the PDF generation system in the Editorial Manager. Otherwise, the figure 2 was submitted to PACE platform according to PLOS One recommendation for Figures. After figure correction using PACE (https://pacev2.apexcovantage.com/), the generated file was submitted with the manuscript. Therefore, we assume that, if accepted for publication, the figure will be more clear on the online version of the paper. A new version of Figure 2 was submitted.

---

## [Editor Report · Decision Letter 1]

24 Nov 2022

Euclasta condylotricha flowers essential oils: a new source of juvenile hormones and its larvicidal activity against Anopheles gambiae s.s. (Diptera: Culicidae)

PONE-D-22-18507R1

Dear Dr. Tchobo,

We’re pleased to inform you that your manuscript has been judged scientifically suitable for publication and will be formally accepted for publication once it meets all outstanding technical requirements.

Kind regards,

Guadalupe Virginia Nevárez-Moorillón, Ph.D.

Academic Editor

PLOS ONE

---

## [Editor Report · Acceptance letter]

13 Jan 2023

PONE-D-22-18507R1 

*Euclasta condylotricha* flowers essential oils: a new source of juvenile hormones and its larvicidal activity against *Anopheles gambiae* s.s. (Diptera: Culicidae) 

Dear Dr. Tchobo:

I'm pleased to inform you that your manuscript has been deemed suitable for publication in PLOS ONE. Congratulations! Your manuscript is now with our production department. 

Kind regards, 

on behalf of

Dr. Guadalupe Virginia Nevárez-Moorillón 

Academic Editor

PLOS ONE